# A Coil-to-Helix Transition Serves as a Binding Motif for hSNF5 and BAF155 Interaction

**DOI:** 10.3390/ijms21072452

**Published:** 2020-04-01

**Authors:** Jeongmin Han, Iktae Kim, Jae-Hyun Park, Ji-Hye Yun, Keehyoung Joo, Taehee Kim, Gye-Young Park, Kyoung-Seok Ryu, Yoon-Joo Ko, Kenji Mizutani, Sam-Young Park, Rho Hyun Seong, Jooyoung Lee, Jeong-Yong Suh, Weontae Lee

**Affiliations:** 1Structural Biochemistry & Molecular Biophysics Laboratory, Department of Biochemistry, College of Life Science and Biotechnology, Yonsei University, Seoul 120-740, Korea; jmhan723@yonsei.ac.kr (J.H.); jhpark@spin.yonsei.ac.kr (J.-H.P.); jihye2@spin.yonsei.ac.kr (J.-H.Y.); thkim@spin.yonsei.ac.kr (T.K.); gypark@spin.yonsei.ac.kr (G.-Y.P.); 2Department of Agricultural Biotechnology and Research Institute of Agriculture and Life Sciences, Seoul National University, 1 Gwanak-ro, Gwanak-gu, Seoul 08826, Korea; iktaekim@snu.ac.kr; 3Center for In Silico Protein Science and Center for Advanced Computation, Korea Institute for Advanced Study, Seoul 130-722, Korea; newton@kias.re.kr; 4Division of Magnetic Resonance Research, Korea Basic Science Institute, Yangcheong-Ri 804-1, Ochang-Eup, Cheongwon-Gun, Chungcheongbuk-Do 363-883, Korea; ksryu@kbsi.re.kr; 5National Center for Inter-University Research Facilities, Seoul National University, 1 Gwanak-ro, Gwanak-gu, Seoul 08826, Korea; yjko@snu.ac.kr; 6Drug Design Laboratory, Graduate School of Medical Life Science, Yokohama City University, Tsurumi, Yokohama 230-0045, Japan; mizutani@yokohama-cu.ac.jp (K.M.); park@yokohama-cu.ac.jp (S.-Y.P.); 7Department of Biological Sciences, Institute of Molecular Biology and Genetics, Research Center for Functional Cellulomics, Seoul National University, Seoul 151-742, Korea; rhseong@snu.ac.kr; 8Center for In Silico Protein Science and School of Computational Sciences, Korea Institute for Advanced Study, Seoul 130-722, Korea

**Keywords:** BAF155, hSNF5, coupled folding and binding, NMR spectroscopy, X-ray crystallography

## Abstract

Human SNF5 and BAF155 constitute the core subunit of multi-protein SWI/SNF chromatin-remodeling complexes that are required for ATP-dependent nucleosome mobility and transcriptional control. Human SNF5 (hSNF5) utilizes its repeat 1 (RPT1) domain to associate with the SWIRM domain of BAF155. Here, we employed X-ray crystallography, nuclear magnetic resonance (NMR) spectroscopy, and various biophysical methods in order to investigate the detailed binding mechanism between hSNF5 and BAF155. Multi-angle light scattering data clearly indicate that hSNF5^171–258^ and BAF155^SWIRM^ are both monomeric in solution and they form a heterodimer. NMR data and crystal structure of the hSNF5^171–258^/BAF155^SWIRM^ complex further reveal a unique binding interface, which involves a coil-to-helix transition upon protein binding. The newly formed α_N_ helix of hSNF5^171–258^ interacts with the β2–α1 loop of hSNF5 via hydrogen bonds and it also displays a hydrophobic interaction with BAF155^SWIRM^. Therefore, the *N*-terminal region of hSNF5^171–258^ plays an important role in tumorigenesis and our data will provide a structural clue for the pathogenesis of Rhabdoid tumors and malignant melanomas that originate from mutations in the *N*-terminal loop region of hSNF5.

## 1. Introduction

Chromatin remodeling is a process that changes the chromatin structure between a condensed state and a transcriptionally accessible state, which is essential for the control of gene expression [1,2,3,4]. The chromatin structures are mediated by various covalent modifications in histones and nucleic acids, along with a set of multi-protein complexes that regulate and respond to these modifications [5]. SWI/SNF (SWItch/Sucrose Non-Fermentable) is an evolutionarily conserved chromatin-remodeling complex that couples ATP hydrolysis with chromatin structure rearrangement. The eukaryotic SWI/SNF complex consists of 12–15 subunits, with an overall molecular weight of ~2 MDa, and it controls diverse cellular processes, such as transcriptional regulation, differentiation, apoptosis, and tumorigenesis [6,7,8,9,10,11,12]. SWI/SNF is comprised of invariable core components, including BRG1/hBRM, SNF5/INI1/BAF47, BAF155/SRG3, and BAF170, as well as a number of variable components that confer functional diversity via combinatorial assembly [13,14,15,16]. The core proteins, SNF5, BAF155, and BAF170, are evolutionarily conserved among yeast, flies, plants, and mammals [17,18]. 

Human SNF5 (hSNF5) regulates the cell cycle at various stages of the mitotic checkpoint and it interacts with a variety of proteins, such as HIV-1 integrase, c-Myc, p53, Epstein-Barr virus nuclear protein 2, and human papillomavirus E1 protein [19,20,21,22]. The interaction with HIV-1 integrase affects viral integration and infectivity [23,24]. hSNF5 is involved in tumor proliferation and progression in he p16-RB pathway, WNT signaling pathway, sonic hedgehog signaling pathway and Polycomb pathway [25]. Additionally, truncating mutations of hSNF5 gene lead to aggressive pediatric atypical teratoid and malignant rhabdoid tumors [25,26]. BAF155 (a human homologue of yeast SWI3) is another core component of the SWI/SNF complex that displays ATPase activity [27,28,29,30]. This protein protects the SWI/SNF complex from proteasomal degradation and it also directs the nuclear localization of the complex [29,31,32]. BAF155 is ubiquitously expressed, being similar to other members of the SWI/SNF complex [33].

Recently, it has been reported that hSNF5 and BAF155 interact via their RPT1 and SWIRM domains, respectively [32,33,34,35]. Here, we investigated the detailed interaction between an *N*- and *C*-terminal elongated hSNF5^RPT1^ (hSNF5^171–258^) domain and the BAF155^SWIRM^ domain while using solution nuclear magnetic resonance (NMR) spectroscopy and X-ray crystallography. Notably, we detected a discrepancy between the binding interface that was identified by NMR-titration experiments and the previous crystal structure (Protein Data Bank (PDB) code 5GJK) [34]. In addition, we found that the crystal structure of the hSNF5^171–258^/BAF155^SWIRM^ complex is dramatically different from that reported in a previous study [34]. Our structure reveals that hSNF5^171–258^ employs its *N*-terminal disordered region as a binding motif for the BAF155^SWIRM^ domain via a coil-to-helix transition. The truncation of the BAF155-binding region from hSNF5 results in a seven-fold decrease in binding affinity, demonstrating the importance of this conformational switch for the hSNF5–BAF155 interaction. Our structure reveals the detailed binding interface between hSNF5 and BAF155 and it highlights a novel folding-upon-binding mechanism in the assembly of this chromatin-remodeling complex.

## 2. Results

### 2.1. hSNF5^171–258^ and BAF155^SWIRM^ form a Heterodimer

hSNF5 is comprised of a winged-helix DNA-binding domain (residues 11–115) and two tandem-repeat domains, repeat 1 (RPT1; residues 186–245) and repeat 2 (RPT2; residues 256–319), which have 39% sequence identity with one another (Figure 1A). Human BAF155 contains five structural domains, including—BRCT, SANT, SWIRM, Glu-rich, and Pro-rich (Figure 1B). Pairwise sequence alignment identity matrices show that the consensus sequence of hSNF5^RPT1^ is highly conserved, whereas the region that extends from the *N*-terminus of hSNF5^RPT1^ exhibits significant sequence variation in distant orthologs (Figure 1C,D). BAF155^SWIRM^ is also well conserved (Figure 1D). We generated BAF155^SWIRM^ (residues 449–546) and eight constructs of hSNF5, including hSNF5^171–258^, hSNF5^171–253^, hSNF5^174–253^, hSNF5^179–253^, hSNF5^181–253^, hSNF5^183–253^, hSNF5^186–253^, and *N*-terminal mutant of hSNF^171–253^ (hSNF5^SFH1/171–253^, a chimeric mutant that contains the *N*-terminal tail region (residues 171–184) from the yeast protein (SFH1)) in this study, to dissect the importance of the region *N*-terminal elongated hSNF5^RPT1^ domain for BAF155 binding (Figure 1A).

A previous report describing the crystal structure of the hSNF5^169–252^/BAF155^SWIRM^ complex found that these proteins form a heterotetramer [34], whereas the size exclusion chromatography (SEC) and multi-angle light scattering (MALS) data have indicated that hSNF5^184–249^ is a monomer [35]. We performed SEC and MALS experiments with different constructs of hSNF5 in complex with BAF155^SWIRM^ to investigate this discrepancy. The calculated molecular weights of hSNF5^171–258^ and BAF155^SWIRM^ were 10.0 kDa and 12.1 kDa, respectively. Using SEC, we found that hSNF5^171–258^ elutes at a higher molecular weight than the monomer fraction, whereas BAF155^SWIRM^ elutes in the monomer fraction (Figure 2A,B). However, MALS data clearly indicate that both hSNF5^171–258^ and BAF155^SWIRM^ are monomeric in solution, with absolute molar masses of 10.0 ± 0.9 kDa and 12.6 ± 1.4 kDa, respectively (Figure 2C,D). The SEC elution profile of hSNF5^171–258^ reflects the effect of a long flexible *N*-terminal loop, as shown in our NMR structure (Figure 2E,F, Table 1). The *N*-terminal region of hSNF5^171–258^ from His171 to Val185 lacks medium- and long-range nuclear Overhauser effects (NOEs) and it remains unstructured in the solution structure of free hSNF5^171–258^. Except for the *N*-terminal loop regions, our solution structure of hSNF5^171-258^ is similar to the previously reported hSNF5^RPT1^ structures covering 184–249 residues (PDB code 5L7A) [35], forming a β1β2α1α2 fold, with a Cα RMSD of 2.0 Å for the 64 atoms between Glu184 and Tyr248 (Figure 2E,F). The hSNF5^171–258^/BAF155^SWIRM^ complex also eluted at an even higher molecular weight than the heterodimer from SEC (Figure 2G), but the absolute molar mass of the hSNF5^171–258^/BAF155^SWIRM^ complex was found to be 22.5 ± 1.6 kDa from MALS data, which confirmed that hSNF5^171–258^ and BAF155^SWIRM^ form a heterodimer (calculated m.w. 22,067 Da) (Figure 2H). Therefore, we clearly defined that hSNF5^171–258^ is a monomer and it binds to BAF155^SWIRM^ in a molar ratio of 1:1, even though their complex eluted in higher molecular weight in the SEC.

### 2.2. An N-Terminal Loop near the hSNF5^RPT1^ Domain Serves as a Binding Motif of BAF155^SWIRM^

Backbone ^1^H, ^15^N, and ^13^C resonances of hSNF5^171–258^ and BAF155^SWIRM^ were assigned while using a suite of three-dimensional (3D) heteronuclear correlation NMR spectroscopy. NMR data for hSNF5^171–258^ and BAF155^SWIRM^ reveal their detailed intermolecular interaction in solution. The chemical-shift perturbation (CSP) of the backbone amide resonances indicates a slow-to-intermediate exchange regime of binding. We assigned backbone resonances of the hSNF5^171–258^/BAF155^SWIRM^ complex from 3D NMR experiments with ^13^C,^15^N-hSNF5^171–258^: BAF155^SWIRM^ and ^13^C,^15^N-BAF155^SWIRM^:hSNF5^171–258^ samples (Figure 3A,B). The heteronuclear single quantum coherence (HSQC) spectra superimposed between the free and complex states of both hSNF5^171–258^ and BAF155^SWIRM^ show that a number of backbone amide resonances are shifted upon interaction. We exploited the backbone chemical shifts of free and complexed proteins in order to obtain CSP diagrams for both hSNF5^171–258^ and BAF155^SWIRM^ upon complex formation (Figure 3C,D). The free form of hSNF5^171–258^ has two β-strands, followed by two α-helices, and BAF155^SWIRM^ is composed of five α-helices, respectively.

Interestingly, our NMR data show that the *N*-terminal region, β2 strand, and β2–α1 loop of hSNF5^171–258^ exhibit significant CSPs, revealing the detailed interaction with BAF155^SWIRM^ (Figure 3C). In particular, the *N*-terminal loop region (His171–Val185) that is near the hSNF5^RPT1^ domain shows a dramatic CSP upon BAF155^SWIRM^ binding. In addition, line widths of the amide resonances of His171, Glu178, Asn179, Ser181, and Gln182 residues in hSNF5^171–258^ are unusually broadened upon BAF155^SWIRM^ binding, which suggested that the *N*-terminal loop region of hSNF5^171–258^ plays an important role in this interaction (Figure 3C). The BAF155^SWIRM^ domain also exhibits CSP in the α4 and α5 helices, as well as in the α4–α5 loop, a finding that is consistent with the previous report [34] (Figure 3D). Interestingly, our data show large CSP in E473 and Q538 in the α2 helix and the *C*-terminal loop of BAF155^SWIRM^, which does not exhibit with the previous structure [34]. Therefore, we expect that the major binding site of hSNF5^171–258^ and BAF155^SWIRM^ during the complex formation differs from the previously reported site. We next determined the X-ray crystal structure of the hSNF5^171–258^/BAF155^SWIRM^ complex to confirm our findings in the solution state for the binding interface between hSNF5^171–258^ and BAF155^SWIRM^.

### 2.3. N-Terminal Loop of hSNF5^RPT1^ Reveals Conformational Change upon BAF155^SWIRM^ Binding

We determined the high-resolution structure of the hSNF5^171–258^/BAF155^SWIRM^ complex using X-ray crystallography for a better understanding of the major binding force between hSNF5^171–258^ and BAF155^SWIRM^. We obtained a viable crystal at conditions of neutral pH and lower ionic strength and then solved the crystal structure of the complex at 2.28 Å resolution (Figure 4A, Table 2). In overall complex structure, hSNF5^171–258^ has an antiparallel β-sheet (β1, 186–195; β2, 198–208), flanked by three α-helices (α_N_, 172–181; α1, 216–226; α2, 230–248), and BAF155^SWIRM^ has five α-helices (α1, 465–470; α2, 472–475; α3, 484–500; α4, 508–514; and, α5, 519–532) (Figure 4A). Although our crystal structure exhibits a similar fold to the previously reported structure (PDB code 5GJK), the additional *N*-terminal helix (α_N_) of hSNF5^171–258^ is only observed in our complex structure (Figure 4A). The electron density map of a newly formed α_N_ helix is clearly observed in our crystal structure and it provides a detailed orientation of the bound conformation (Figure 4B). The α_N_ helix and β2–α1 loop in hSNF5^171–258^ intramolecularly interacts via hydrogen bonds and hydrophobic interaction following the residues, such as Val175, Asn179, Lys211, and Leu212 (Figure 4C). Strikingly, in our complex structure, their secondary structure of the *N*-terminal loop of hSNF5^171–258^, which shows the large CSP change during BAF155^SWIRM^ binding, dramatically differs from its free state structure showing the random-coil fold (Figure 2E,F). We note that the conformational change mainly occurs in the *N*-terminal loop region, otherwise hSNF5^171–258^ in complex maintains the same structure as free hSNF5^171–258^ in solution, yielding a Cα RMSD of 2.2 Å for secondary structural regions. Therefore, we found that there is a conformational transition from the coil to the helix in the *N*-terminal loop of hSNF5^171–258^ during the complex state with BAF155^SWIRM^ (Figure 4D).

^1^H-^15^N heteronuclear NOE (XNOE) data for hSNF5^171-258^ in solution indicates the overall structural flexibility. The compactly folded secondary structure parts, β1β2α1α2, reveals high XNOE values, whereas the *N*-terminal and *C*-terminal regions exhibited reduced XNOE values, which indicated a flexible and disordered conformation (Figure 4E). We could not observe the XNOE values of all the participating residues binding with BAF155^SWIRM^ because of the line broadening. However, we find a valid increment of the XNOE values in the residues from Asp172 to His177 locating the *N*-terminal helix after it binds with BAF155^SWIRM^. It should also be noted that the *C*-terminal region (residues 249–258) of hSNF5^171–258^ lacks a regular secondary structure in both the free state and the complex (Figure 4D,E).

The combined data from the NMR spectroscopy and X-ray crystallography uncovered that the *N*-terminal region of hSNF5^171–258^ undergoes a coil-to-helix transition upon its binding with BAF155^SWIRM^, illustrating a coupled folding and binding mechanism. 

### 2.4. The Interface between hSNF5^171–258^ and BAF155^SWIRM^ Features Charge Complementarity

hSNF5^171–258^ is a highly acidic protein (pI = 4.1), whereas BAF155^SWIRM^ is basic (pI = 7.9). The inspection of surface charges at the binding interface reveals charge complementarity, such that positively charged surfaces of BAF155^SWIRM^ (Appendix A) interact with negatively charged surfaces of hSNF5^171–258^ (Appendix A). Complex formation between hSNF5^171–258^ and BAF155^SWIRM^ buries 1684 Å2 of solvent-accessible surface area, which is ~30% larger than the buried surface area of 1308 Å2 detected in the same complex without the α_N_ helix. The interaction surface features key salt bridges and hydrogen bonds between hSNF5^171–258^ and BAF155^SWIRM^ (Asp202—Arg512/Arg513 and Glu210—Arg524) (Figure 5A). In addition, the Asn207 of hSNF5^171–258^ forms a hydrogen bond to Asp518 of BAF155^SWIRM^, and Asp225 of hSNF5^171–258^ forms a hydrogen-bonding network with Ser508/Thr509 of BAF155^SWIRM^. The binding interface with hSNF5^171–258^ and BAF155^SWIRM^ is similar to that of the previous structure [34], but the binding distance between the residues are quite different (Figure 5A). Specifically, the salt bridge distance between Asp202 and Arg512 in our hSNF5^171–258^/BAF155^SWIRM^ complex structure is about 0.7Å further apart when compared with that of the previous structure [34]. On the other hand, the salt bridge distance between Glu210 and Arg524 in our hSNF5^171–258^/BAF155^SWIRM^ complex structure is approximately 0.3Å closer than that of the previous structure (Figure 5A). Moreover, we also found an additional hydrogen bond network in the presence of the α_N_ helix of hSNF5^171–258^. A hydrogen bond network formed between the Asn179 of hSNF5^171–258^ and Arg524 and the Glu473 of BAF155^SWIRM^. The Asn179 located in the α_N_ helix of hSNF5^171–258^, showing a high CSP value, also interacts directly with Lys211 (Figure 3C and Figure 5B). Therefore, it is considered that the newly observed hydrogen bond network that is mediated by the Asn179 in the α_N_ helix of hSNF5^171–258^ is a very important binding force for the hSNF5^171–258^/BAF155^SWIRM^ complex as well as hydrogen and hydrophobic interactions that were observed on the charged surface (Figure 5B). In the NMR titration result, the Glu473 of BAF155^SWIRM^ represents a high CSP value, which is consistent with our hSNF5^171–258^/BAF155^SWIRM^ complex structure (Figure 3D and Figure 5B). In addition, hydrogen bond pairs (Ala180—Asn476, Gln182—Asn479, and Glu184—Ans479/Ser481) form in a complex formation of hSNF5^171–258^/BAF155^SWIRM^, which makes hydrogen bonding networks in the *N*-terminal region of hSNF5^171–258^. Backbone carbonyl groups of Ala180 and Gln182 from hSNF5^171–258^ form hydrogen bonds with side chains of Asn476 and Asn479, respectively, from BAF155^SWIRM^. The carboxyl side chain of Glu184 from hSNF5^171–258^ forms hydrogen bonds and salt bridges with the side chains of Ser481 and Lys482 of BAF155^SWIRM^ (Figure 5C). The α_N_ helix also induces some hydrophobic interactions. Three residues, Ile176, His177, and Ala180, provide the hydrophobic interfaces for Pro472 of BAF155^SWIRM^.

The crystal structure in this study is fully consistent with the CSP analysis from the NMR-titration experiments. Finally, we identified that the *N*-terminal region of hSNF5^171–258^ has a conformational change and it induces a tight binding pocket during the complex state.

### 2.5. The N-Terminal Binding Motif of hSNF5^171–258^ Enhances the Molecular Interaction for Stabilizing hSNF5^171–258^/BAF155^SWIRM^ Complex

We further investigated how the *N*-terminal region of hSNF5^171–258^ is important for the target binding and stability of the protein complex. Data from isothermal titration calorimetry (ITC) experiments support the importance of the α_N_ helix for BAF155^SWIRM^ binding in a quantitative manner. We observed that hSNF5^171–258^ and hSNF5^171–253^ exhibit the same binding affinity to BAF155^SWIRM^, which indicates that the five residues in the *C*-terminal tail do not affect protein binding. We also prepared a series of *N*-terminal truncation hSNF5^171–253^ mutants and measured how these truncations impact BAF155^SWIRM^ binding (Table 3). Our data show that hSNF5^171–253^ binds to BAF155^SWIRM^, with a dissociation constant (*K*_D_) of 110 ± 20 nM and a 1:1 stoichiometry (Figure 6A). 

The binding affinity reduced by 2.6-fold when the whole α_N_ helix was removed (hSNF5^183–253^) (KD ~290 ± 80 nM). The moderate decrease in affinity suggests that the α_N_ helix might engage with the BAF155^SWIRM^ via transient interactions. We note that the α_N_ helix exhibited relatively low heteronuclear NOE (XNOE) values, even in complex with the BAF155^SWIRM^, as shown in Figure 4E. Therefore, the α_N_ helix in the hSNF5^171-258^/BAF155^SWIRM^ complex in solution could retain a degree of conformational mobility, providing favorable interactions for stabilizing the complex.

Progressive truncations of the α_N_ helix show that hSNF5^174–253^ maintains the same binding affinity, with a *K*_D_ of ~110 ± 30 nM, whereas hSNF5^179–253^ and hSNF5^183–253^ show decreased binding affinities, with *K*_D_ values of ~300 ± 50 nM and ~290 ± 80 nM, respectively, similar to that observed for the whole α_N_ helix-deletion mutant. The truncation of the *N*-terminal loop preceding the β1 strand results in a ~7-fold reduction in binding affinity, such that hSNF5^186–253^ exhibits a *K*_D_ of ~758 ± 80.3 nM for BAF155^SWIRM^ binding (Fig. 6B). This large reduction results from the loss of hydrogen bonding from Glu184 of hSNF5^171-253^, as well as the α_N_-helix deletion. We note that a single mutation of Glu184 reduces the binding affinity by 3.4-fold, yielding a *K*_D_ of ~370 ± 100 nM for the E184A mutation of hSNF5^171–253^ (Figure 6C). We further found that alanine mutations of Asn476 and Asn479 in BAF155^SWIRM^, which make hydrogen bonds to Ala180 and Gln182, respectively, in hSNF5^171–253^ have a negligible impact on binding. The N476A/N479A mutant of BAF155^SWIRM^ only shows a modest decrease in binding affinity, with a *K*_D_ of ~140 ± 30 nM, indicating a minor role or replaceable nature for their interactions. Finally, the *N*-terminal mutant of hSNF^171–253^ (hSNF5^SFH1/171–253^), followed by the human sequence (residues 185–253), exhibits a 4.4-fold lower binding affinity. The *N*-terminal tail region that extends 15 amino acids in the RPT1 sequence, is not conserved between the human and yeast sequences, so this reduced binding results from divergent sequences in the α_N_ helix and its surrounding region.

Lastly, we investigated whether the α_N_ helix formation significantly contributes to the stability of the hSNF5^171–253^/BAF155^SWIRM^ complex by measuring the melting temperature (Tm) with circular dichroism (CD) spectroscopy. Although hSNF5^171–253^ and hSNF5^186-253^ both exhibit the same Tm of 49 °C in the free state, hSNF5^171–253^ in complex has a higher Tm (~53 °C) than hSNF5^186–253^ (Tm ~48 °C), demonstrating that the α_N_ helix is a determinant of protein stability for the hSNF5^171–253^/BAF155^SWIRM^ complex (Figure 6D,E).

## 3. Discussion

The core complex that is composed of SNF5, BAF155, and BAF170 was found to show a remodeling activity that resembles that of the entire mammalian SWI/SNF complex [38,39]. An important step toward understanding how specific subunits function is to elucidate how they interact within the complex. To date, an atomic-resolution structure of the human SWI/SNF complex has not been obtained, although several individual subunits and domains have been described. Here, we solved the structure of the hSNF5^171–258^/BAF155^SWIRM^ complex at atomic resolution, providing biophysically and functionally important information. Our data indicate that the *N*-terminal loop preceding hSNF5^RPT1^ undergoes a conformational change upon interaction with BAF155^SWIRM^, which enhances the binding strength and stability of the complex state. The yeast Cryo-EM structure showed the presence of α_N_ helix in the SWI/SNF complex [40]. Further, the hydrogen bond network and hydrophobic interactions between the α_N_ helix and the SWIRM domain in our structure recapitulated, illustrating that the α_N_ helix presents the bona fide interface for the SWIRM domain.

Interestingly, a number of studies report that mutations in the *N*-terminal residues Pro173 and Glu184 of hSNF5 lead to different cancers, such as Rhabdoid tumors and malignant melanoma [41,42]. The Glu184 residue of hSNF5^171–258^ plays an important role in binding between the *N*-terminal region of hSNF5^171–258^ and BAF155^SWIRM^, and it is involved in tumor suppression [41]. We speculate that the mutation of Glu184, where the *N*-terminal region of hSNF5^171–258^ binds to BAF155^SWIRM^, interrupts the formation of the hSNF5 and BAF155 complex, making it difficult to form chromatin-remodeling complexes and chromatin, and thereby resulting in tumor genesis. When considering that mutations of Glu184 and Val185 abolish the interaction with HIV-1 integrase (IN) [43], we infer that residues on the non-structured loop following the *N*-terminal tail play an important role in the interaction with IN to a greater extent than with BAF155^SWIRM^. 

Intrinsically disordered regions can perform various biological functions via coupled folding and binding mechanisms [44,45,46,47,48]. The complex structure of hSNF5^171–258^ and BAF155^SWIRM^ illustrates that the interaction between hSNF5 and BAF155 also involves a coupled folding and binding mechanism. The coil-to-helix transition of hSNF5^171–258^ upon BAF155^SWIRM^ binding might help to distinguish between various targets, thereby fine-tuning individual interactions. The structural transition of hSNF5^171–258^ can alter chromatin states and remodeling activity in vivo for the temporal and spatial control of biological processes, such as the recruitment of chromatin remodelers or transcriptional activation.

In summary, we have determined the crystal structure of the hSNF5^171–258^/BAF155^SWIRM^ complex as a heterodimer by 1:1 molar ratio, which reveals the detailed binding mode. The *N*-terminal disordered loop of hSNF5^171–258^ undergoes a coil-to-helix transition upon binding to BAF155^SWIRM^. In the *N*-terminal region of hSNF5^171–258^, residues between Asn179 and Glu184 were identified as core regions that bind to BAF155^SWIRM^, which is critical for protein binding and it enhances the stability of the protein complex. In addition, our data will provide a structural clue to understanding the mechanism of the pathogenesis of diseases that are related to the chromatin remodeling complex and originated from mutations in the *N*-terminal region of hSNF5.

## 4. Materials and Methods 

### 4.1. Cloning, Protein Expression, and Purification

The cDNA fragments encoding residues 171–258 of hSNF5 (hSNF5^171–258^) and the various *N*-terminal truncation mutants and residues 449–546 of human BAF155 (BAF155^SWIRM^) and the BAF155^SWIRM^ mutants were amplified by PCR. The amplified cDNA fragments were inserted into the modified expression vector pET21b (Novagen, Darmstadt, Germany) for hSNF5^171–258^ and the *N*-terminal truncation mutants and pMAL-C2X (Novagen, Darmstadt, Germany) for BAF155^SWIRM^ and the BAF155^SWIRM^ mutants as fusions with *N*-terminal His_6_-affinity tags and tobacco etch virus (TEV) protease cleavage sites. The plasmids were transformed into Escherichia coli Strain BL21 (DE3) (Invitrogen, Waltham, MA, USA). Cells expressing hSNF5^171-258^ and BAF155^SWIRM^ were grown in LB or M9 minimal media with ^15^NH_4_Cl and/or ^13^C_6_-glucose as the sole nitrogen and carbon sources, respectively, at 37 °C to an OD_600_ of ~0.6. Protein expression was induced with 1 mM isopropyl-β-d-thiogalactopyranoside (IPTG) at 25 °C and the cells were harvested by centrifugation after 16 h of induction and then stored frozen at −70 °C. Cell pellets were resuspended and disrupted by sonication in lysis buffer containing 25 mM sodium phosphate, pH 7.0, 300 mM NaCl, 5 mM β-mercaptoethanol, and protease inhibitor cocktail (Roche, Basel, Switzerland). The His_6_-tagged fusion proteins were purified by immobilized metal affinity chromatography on a Ni-NTA column (QIAGEN, Hilden, Germany) and then cleaved by TEV protease for 12 h. The digestion reactions were loaded onto a Ni-NTA column, and the flow-through was collected and loaded onto a Superdex 200 16/60 column (GE Healthcare, Chicago, IL, USA), equilibrated with 10 mM HEPES, pH 7.0, 100 mM NaCl, and 2 mM dithiothreitol (DTT).

### 4.2. Size Exclusion Chromatography and Multi-Angle Light Scattering Analysis

SEC analysis was performed on a Superdex 200 16/60 column (GE Healthcare, Chicago, IL, USA) at 25 °C. The column was equilibrated with buffer containing 10 mM HEPES, pH 7.0, 100 mM NaCl, and 2 mM DTT. Purified hSNF5^171-258^ (0.2 mM) and BAF155^SWIRM^ (0.3 mM) were loaded onto the column, individually or together. The following molecular weight marker proteins were used: aldolase (158 kDa), conalbumin (75 kDa), ovalbumin (44 kDa), carbonic anhydrase (29 kDa), ribonuclease A (13.7 kDa), and aprotinin (6.5 kDa). The molecular weights of free protein and protein-protein complexes were calculated while using the following equation: Y = −0.162X + 2.2157, where R^2^ = 0.9955, and X = V_e_/V_o_ (V_o_ = 45.15 mL, V_e_ = elution volume, Y = logM_W_). Sodium dodecyl sulfate-polyacrylamide gel electrophoresis (SDS-PAGE) confirmed the eluted samples.

SEC multi-angle light scattering (SEC-MALS) experiments were performed using the Agilent 1200 HPLC system (Agilent Technologies, Santa Clara, CA, USA), combined with a Wyatt DAWN HELEOS-II MALS instrument and a Wyatt OPtilab rEX differential refractometer (Wyatt Technology, Santa Barbara, CA, USA). For chromatographic separation, a WTC-030S5 column (Wyatt Technology, Santa Barbara, CA, USA) was used at a flow rate of 0.5 mL/min. in the same buffer as was used for SEC experiments. The same concentrations of hSNF5^171-258^, BAF155^SWIRM^, and hSNF5^171–258^/BAF155^SWIRM^ complex that were used for SEC analysis were also used in the SEC-MALS experiments. The results were normalized and analyzed while using ASTRA software (Wyatt Technology, Santa Barbara, CA, USA).

### 4.3. NMR Spectroscopy

The NMR samples contained 0.5 mM ^13^C,^15^N-hSNF5^171–258^ or ^13^C,^15^N-BAF155^SWIRM^ in 10 mM HEPES, pH 7.0, 100 mM NaCl, and 2 mM DTT. The NMR spectra were recorded at 25 °C on a Bruker 500 and 800 MHz spectrometer, which was equipped with a z-shielded gradient triple resonance probe. Sequential assignments of ^1^H, ^15^N, and ^13^C resonance were performed by 3D triple resonance through-bond scalar correlation experiments (CBCACONH, HNCACB, HNCO, HNCACO, HNHA, (H)CCC(CO)NH, and H(CCCO)NH). For structure calculation purposes, ^15^N-edited NOESY (τm = 150 ms) and ^13^C-edited NOESY-HSQC (τm = 120 ms) experiments were acquired. Dihedral backbone restraints were derived from ^1^Ha, ^13^Ca, ^13^Cb, and ^13^CO chemical shifts with the TALOS program [49]. The residual ^1^D_NH_ dipolar couplings were obtained by taking the difference in the J splitting values measured in oriented (12 mg/mL of pf1 alignment media) and isotropic (water) hSNF5^171-258^ using 2D in-phase/antiphase ^1^H-^15^N HSQC spectra [49]. Heteronuclear ^1^H-^15^N NOE was measured using pulse schemes, as described previously [50]. The NMR spectra were processed and analyzed using the NMRPipe [51] and NMRFAM-SPARKY [52] programs. ^15^N-HSQC titration experiments were performed at 1:1molar ratios of ^15^N-labeled proteins to unlabeled counter proteins between hSNF5^171-258^ and BAF155^SWIRM^. Chemical-shift perturbations were calculated using the equation Δδ_AV_ = ((Δδ_1H_)^2^ + (Δδ_15N_/5)^2^)^1/2^, where Δδ_AV_, Δδ_1H_, and Δδ_15N_ are the average, proton, and ^15^N chemical-shift changes, respectively.

### 4.4. NMR Structure Determination and Analysis

The structure calculations for BAF155^SWIRM^ and hSNF5^171–258^ in the free state were performed using semi-automated CYANA 2.1 [53]. The initial fold was obtained by manual NOE assignment, and secondary topology was determined by peak picking. NOE cross-peak assignments were obtained using manual procedures. An initial fold of the protein was calculated on the basis of manually assigned NOEs, which is essential for secondary topology determination, with subsequent use of the program CYANA 2.1. Seven cycles of the CYANA routine were calculated; each cycle had 10,000 steps of torsion-angle dynamics, with a simulated annealing protocol. A total of 100 structures were calculated, and the 20 structures with the lowest target function values were chosen for analysis. Table 2 shows a summary of the NMR-derived restraints and structures of hSNF5^171–258^. The final structures with the lowest NOE energies were retained and validated with the PROCHECK program [54]. The structures were analyzed and visualized using PyMOL (Delano Scientific LLC, San Carlos, CA, USA, https://pymol.org/) and MOLMOL(https://sourceforge.net/projects/molmol/) [55].

### 4.5. Crystallization and Structure Determination

Crystals of the hSNF5^171–258^/BAF155^SWIRM^ complex were grown under oil at 15 °C in a 1.5 µL micro-batch, containing equal volumes of protein solution and mother liquor (15% *v*/*v* Tacsimate, pH 7.0, 0.1 M HEPES, pH 7.0, 2% (*w*/*v*) polyethylene glycol 3350). Purified hSNF5^171–258^ was mixed with BAF155^SWIRM^ at a 1:1 molar ratio. The crystals were flash-cooled in liquid nitrogen, with 25% glycerol being used as a cryoprotectant. The diffraction datasets were collected using a synchrotron radiation source at beamline 4A at the Pohang Accelerator Laboratory (Pohang, Korea) at 100 K. Collected data were integrated and scaled while using the HKL-2000 program (HKL Research Inc., Charlottesville, VA, USA). The crystal structure of the hSNF5^171–258^/BAF155^SWIRM^ complex was solved by the molecular replacement method with the MOLREP [56] and Phaser programs, using a high-accuracy template-based modeling method as the search model. The final structure was refined to the maximum resolution of 2.28 Å by iterative cycles of the manual building in COOT and then restrained refinement with Refmac5 and Phenix [57]. Table 1 summarizes data collection and refinement for the complex structure. The final structures were analyzed while using MolProbity [58].

### 4.6. Crystallization and Structure Determination

The ITC experiments were performed using a VP-ITC system (MicroCal Inc. Northampton, MA, USA) at 25 °C in a buffer containing 25 mM sodium phosphate, 100 mM NaCl, and 1 mM Tris(2-carboxyethyl)-phosphine (TCEP), pH 7.0. Typically, 160 µM hSNF5^171–253^ was injected 35 times in 7 µL aliquots into the 1.4 mL sample cell, containing either the BAF155^SWIRM^ domain or BAF155^SWIRM^ mutants at a concentration of 10 μM. The data were fit with a non-linear least-squares best-fit curve, while using a single-site binding model with Origin for ITC (Microcal Inc.), by varying the stoichiometry (*n*), the enthalpy of the reaction (ΔH), and the dissociation constant (*K*_D_).

### 4.7. Circular Dichroism Spectroscopy

CD experiments to measure the melting temperatures were performed in buffer containing 25 mM sodium phosphate, 100 mM NaCl, and 1 mM Tris(2-carboxyethyl)-phosphine (TCEP), pH 7.0, as described previously [59,60]. Samples containing 10 μM hSNF5^171–253^ or hSNF5^186–253^ alone or with 10 µM BAF155^SWIRM^ were monitored at temperatures between 25 °C and 95 °C, with an interval of 5 °C. The parameters for far-ultraviolet UV CD measurements used a cell with a path length of 0.2 cm for scanning between 250–200 nm, with a 1 nm bandwidth and scan speed of 200 nm/min^−1^. The values were baseline corrected by subtracting a buffer spectrum, and an average of five scans was recorded for each experiment. The apparent melting temperature at the midpoint of the transition was obtained by fitting the experimental data points (CD signal at 222 nm vs. temperature) with a Boltzmann sigmoidal function.

## Figures and Tables

**Figure 1 ijms-21-02452-f001:**
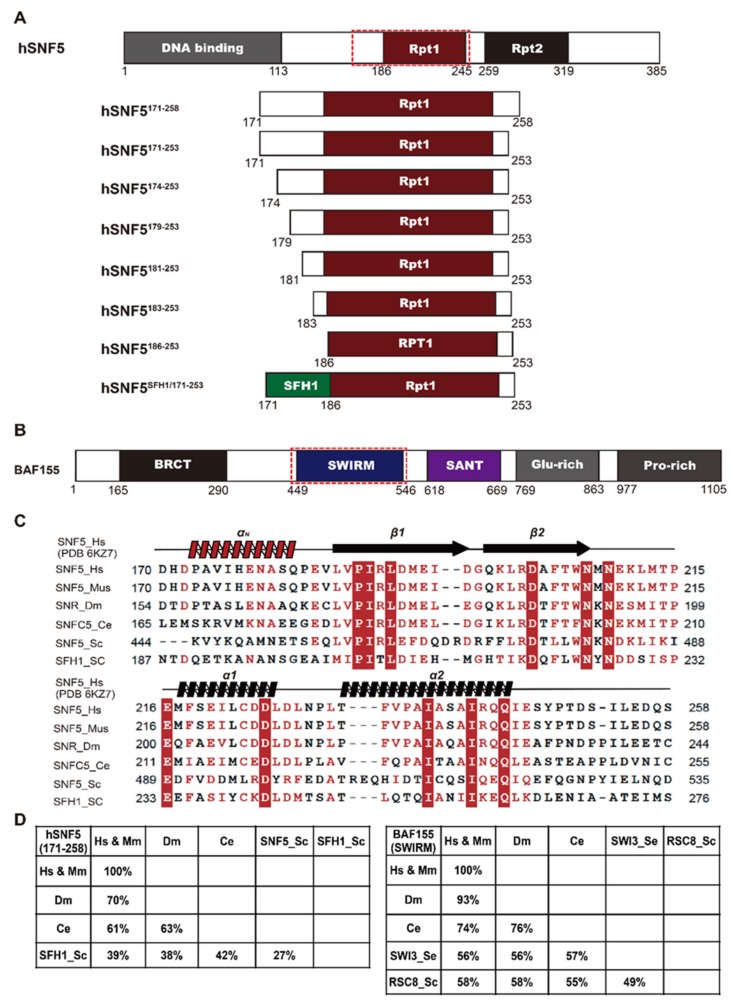
Domain structures of hSNF5 and BAF155. (**A**) Schematic representation of full-length hSNF5 and eight hSNF5 constructs containing the hSNF5^RPT1^ domain. (**B**) Domain structure of full-length BAF155. The hSNF5^RPT1^ (residues 186–245) domain (red) and the BAF155^SWIRM^ domain (blue); red dotted box indicates the interacting region. (**C**) Multiple sequence alignment of the hSNF5^171–258^ region with homologs in mouse, fruit fly, nematode, and yeast. Sequence alignment was performed using T-coffee software [36] and rendered further with ESPript software [37]. (**D**) Pairwise sequence alignment identity matrices for hSNF5^171–258^ (left) and BAF155^SWIRM^ (right) and their homologs and orthologs. Hs, Homo sapiens; Mm, Mus musculus; Dm, Drosophila melanogaster; Ce, Caenorhabditis elegans; and Sc, Saccharomyces cerevisiae. For both hSNF5^171–258^ and BAF155^SWIRM^, the sequence in Hs is identical to that in Mm.

**Figure 2 ijms-21-02452-f002:**
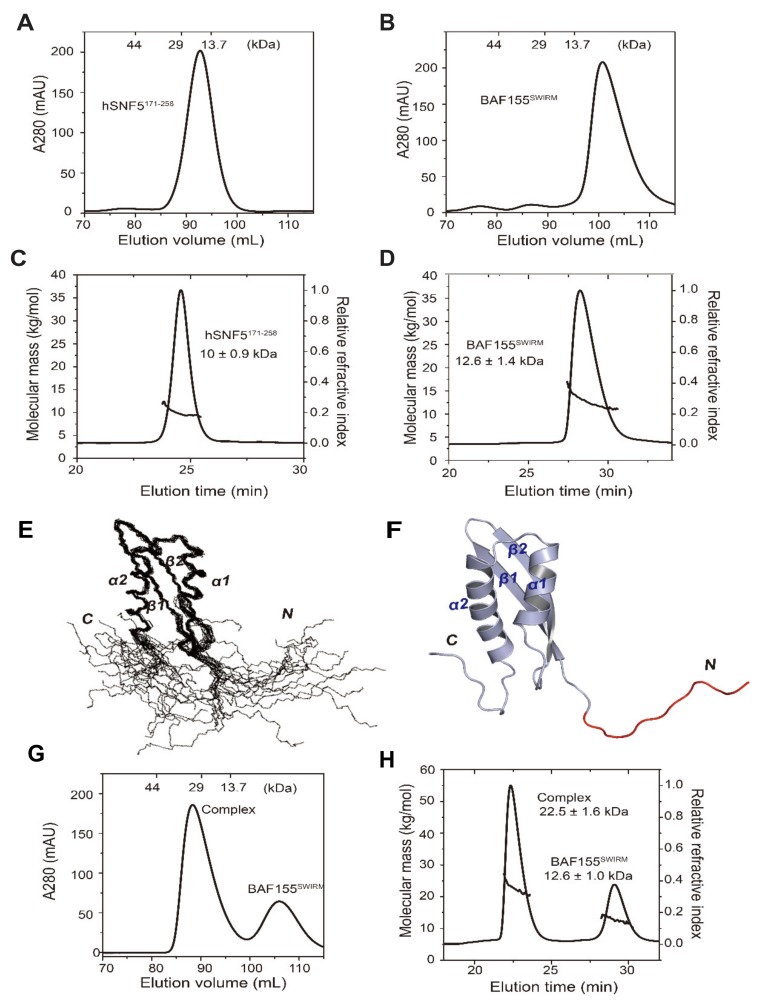
Analysis of the complex formation of hSNF5^171–258^, binding with BAF155^SWIRM^, and its solution structure. Size exclusion chromatography (SEC) analysis using a Superdex 200 16/60 column for (**A**) hSNF5^171–258^ (~13.7 kDa) and (**B**) BAF155^SWIRM^ (~13 kDa). SEC-multi-angle light scattering (MALS) elution profiles from a Wyatt WTC-030S5 column for (**C**) hSNF5^171–258^ and (**D**) BAF155^SWIRM^. (**E**) Superposition of the backbone atoms of the final 20 lowest-energy calculated structure and (**F**) a ribbon model of hSNF5^171–258^. The *N*-terminal region of hSNF5^171–258^ is shown in the flexible loop (red). (**G**) SEC profile of the hSNF5^171–258^/BAF155^SWIRM^ complex (~29 kDa). Elution profiles of standard marker protein are shown at the top of the chromatograms as a reference. (**H**) SEC-MALS elution profiles of the hSNF5^171–258^/BAF155^SWIRM^ complex. The molecular masses obtained from light scattering and refractive index measurements correspond to the molecular weights of free hSNF5^171–258^, free BAF155^SWIRM^, and the hSNF5^171–258^/BAF155^SWIRM^ complex.

**Figure 3 ijms-21-02452-f003:**
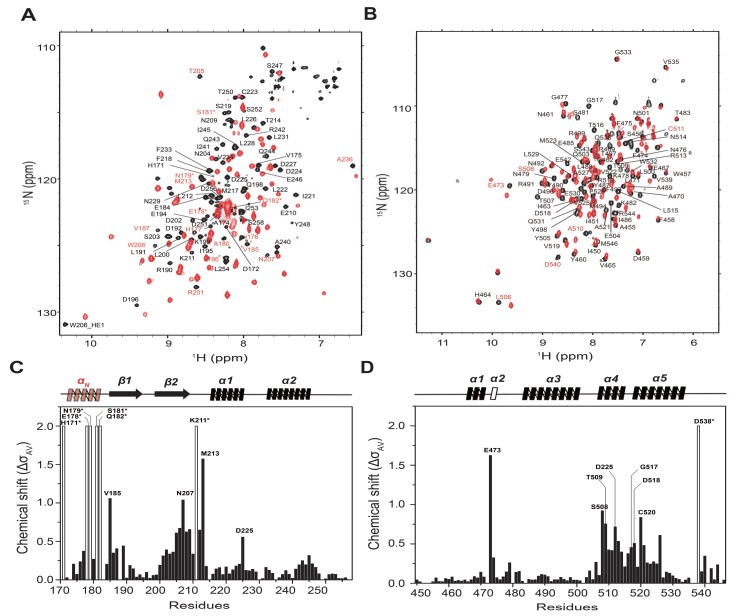
The interaction site mapping on hSNF5^171–258^/BAF155^SWIRM^ complex using nuclear magnetic resonance (NMR) titration. (**A**) ^1^H-^15^N heteronuclear single quantum coherence (HSQC) spectrum of free ^15^N-hSNF5^171–258^ (black) was superimposed with the spectrum of ^15^N-hSNF5^171–258^/BAF155^SWIRM^ complex (red). (**B**) ^1^H-^15^N HSQC spectrum of free ^15^N-BAF155^SWIRM^ (black) was superimposed with the spectrum of the ^15^N-BAF155^SWIRM^/hSNF5^171–258^ complex (red). For these experiments, 0.3 mM 15N-hSNF5^171–258^ or ^15^N-BAF155^SWIRM^ was stoichiometrically titrated with an equivalent amount of unlabeled BAF155^SWIRM^ or hSNF5^171–258^, respectively. The residues that show a large chemical-shift perturbation (CSP) are annotated. CSP plots for (**C**) hSNF5^171–258^ and (**D**) BAF155^SWIRM^, as a function of the residue number obtained from the NMR titration. Average chemical-shift changes were calculated using the following formula: Δδ_AV_ = ((Δδ_1H_)^2^ + (Δδ_15N_/5)^2^)^1/2^. Secondary structures are shown above the CSP plots. Residues with broadened line widths are indicated with stars (*), and the secondary structure shown in transparent red represents the α-helix in the *N*-terminal region of hSNF5^171–258^ (α_N_) formed upon binding with BAF155^SWIRM^.

**Figure 4 ijms-21-02452-f004:**
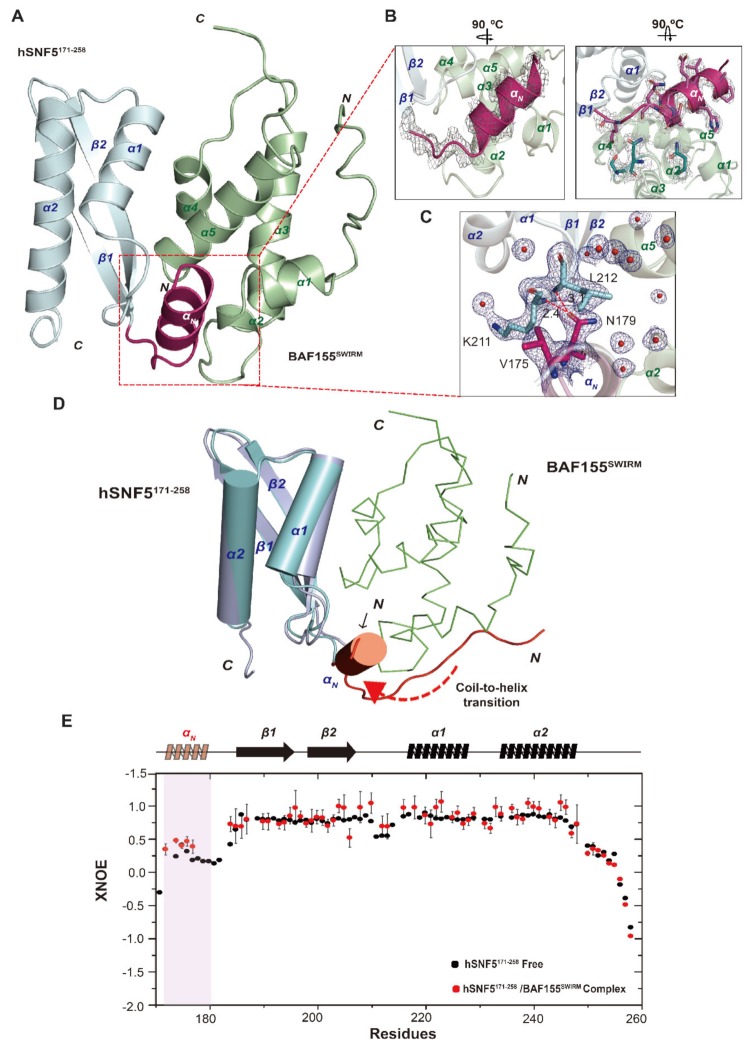
Conformational changes of the *N*-terminal region of hSNF5^171–258^ during the complex formation with BAF155^SWIRM^. (**A**) The high-resolution crystal structure of hSNF5^171–258^/BAF155^SWIRM^ complex is shown as a cartoon diagram. Both molecules, hSNF5^171–258^ and BAF155^SWIRM^, are colored cyan and green, respectively. Secondary structures are annotated, and the *N*-terminal region of hSNF5^171–258^ is shown in pink. (**B**) The *N*-terminal helix region of the hSNF5^171–258^ is plotted while using an omit map at the contour level 1.2 σ as a gray mesh and it shows both the backbone (left) and the side chain (right) electron density map. (**C**) The intermolecular hydrophobic and hydrogen bond interactions between α_N_ helix and β2–α1 loop in hSNF5^171–258^ are depicted as sticks and blue mesh. Hydrogen bonds and water molecules are shown in the red dashed lines and red sphere, respectively. (**D**) The solution structure of hSNF5^171–258^ in a free state (purple ribbon model) is superimposed with the crystal structure of hSNF5^171-258^/BAF155^SWIRM^ in a complex state (cyan ribbon model). The *N*-terminal region of hSNF5^171–258^ in both structures, that experience the conformational change from the coil to helix, are marked by red colors, respectively. (**E**) Backbone dynamics profile of hSNF5^171–258^ in a free state (black dot) and the BAF155^SWIRM^ bound state (red dot). The XNOE data were plotted against the residue number and the corresponding secondary structure of hSNF5^171–258^. A red box indicates the *N*-terminal region of hSNF5^171–258^, forming helix upon binding with BAF155^SWIRM^, and a red transparent helix represents the secondary structure.

**Figure 5 ijms-21-02452-f005:**
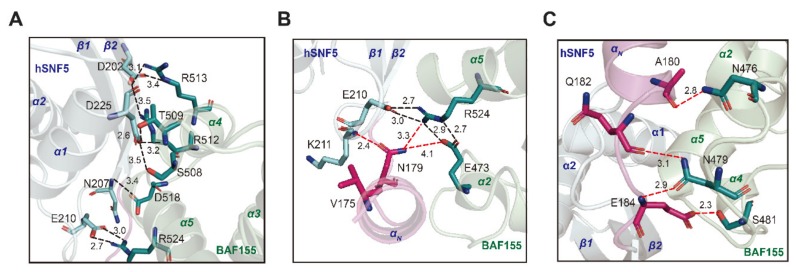
Detailed view of residual networks against with the *N*-terminal helix of hSNF5^171–258^ complexed with BAF155^SWIRM^. (**A**) The hydrogen bond network in the binding interface between hSNF5^171–258^ (cyan) and BAF155^SWIRM^ (green). (**B**,**C**) The additional hydrogen bond network mediated by the *N*-terminal region (magenta) of hSNF5^171–258^ in complex with BAF155^SWIRM^ is shown in stick model. The residual networks newly represented by the *N*-terminal loop are marked with a red dash to distinguish it from the known interacting network.

**Figure 6 ijms-21-02452-f006:**
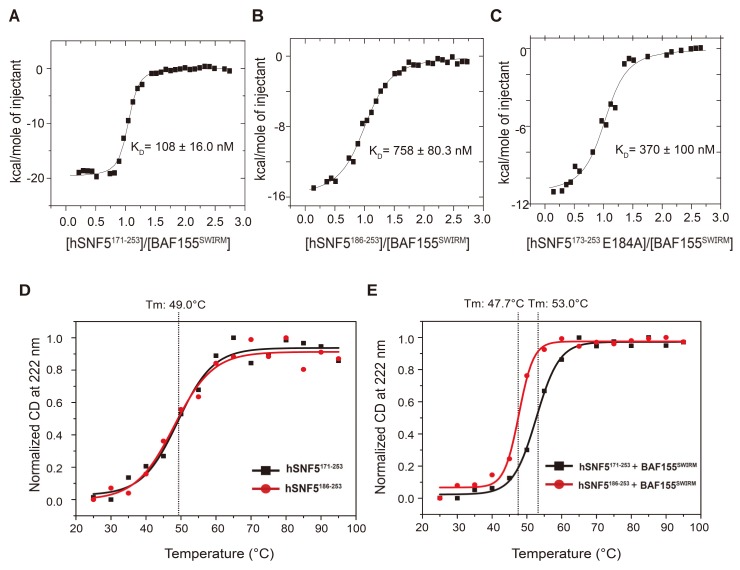
*N*-terminal disordered regions of hSNF5^171–253^ contribute to the enhancement of the interaction with BAF155^SWIRM^. Isothermal titration calorimetry (ITC) for the interaction between (**A**) hSNF5^171–253^ and BAF155^SWIRM^, (**B**) hSNF5^186–253^ and BAF155^SWIRM^, and (**C**) hSNF5^171-253^ with the E184A mutation and BAF155^SWIRM^. The integrated heat of injection is presented, and the experimental data are shown as solid squares. The least-squares best-fit curves derived from a simple one-site binding model are shown as a black line. Thermodynamic parameters are shown in Table 3. Thermal stability of hSNF5^171–253^ and hSNF5^186–253^ (**D**) alone and (**E**) with BAF155^SWIRM^. The melting temperatures (Tm) were calculated by the decrease in normalized molar ellipticity at 222 nm in the circular dichroism (CD) spectrum.

**Table 1 ijms-21-02452-t001:** Restraints and structural statistics for hSNF5^171–258^.

Experimental Restraints	<SA> *
Non-redundant NOEs	1891
Dihedral angles, φ/ψ	62/62
Hydrogen bonds	25
Residual dipolar coupling, ^1^D_NH_	77
Total number of restraints	1782 (24.3 per residue)
RMSD from experimental restraints	
Distances (Å) (1891)	0.034 ± 0.002
Torsion angles (°) (124)	2.34 ± 0.14
Residual dipolar coupling *R*-factor (%) ^†^	
^1^D_NH_ (%) (77)	2.9 ± 0.4
RMSD from idealized covalent geometry	
Bonds (Å)	0.003 ± 0
Angles (°)	0.59 ± 0.01
Impropers (°)	0.67 ± 0.02
Coordinate precision (Å) *^‡^	
Backbone	0.48 ± 0.15
Heavy atoms	1.29 ± 0.17
Ramachandran statistics (%) ^‡§^	
Most favorable regions	80.5 ± 1.1
Allowed regions	19.5 ± 1.1

* For the ensemble of the final 20 simulated annealing structures. ^†^ The magnitudes of the axial and rhombic components of the alignment tensor were 5.4 Hz and 0.65, respectively. ^‡^ Regions with secondary structures (residues 186−195, 198−207, 218−225, and 233−247). ^§^ Calculated using the program PROCHECK.

**Table 2 ijms-21-02452-t002:** Data collection and refinement statistics for the hSNF5^171–258^/BAF155^SWIRM^ complex.

hSNF5^171–258^/BAF155^SWIRM^ Complex	
**Data Collection**	
Space group	*H3*
Cell dimensions	
*a, b, c* (Å)	77.23, 77.23, 207.24
*α, β, γ* (°)	90.00, 90.00, 120.00
Resolution (Å)^a^	28.1–2.28 (2.236–2.28)
*I/σI*	20.16 (3.01)
*R*_merge_ (%) ^a^	8.5 (40.5)
Completeness (%) ^a^	99.87 (99.9)
Redundancy ^a^	5.43 (4.8)
**Refinement**	
Resolution (Å)	28.1–2.28 (2.236–2.28)
No. reflections	20987
*R*_work_/*R*_free_	0.1671/0.2003
No. atoms	
Protein	2876
water	223
*B*-factors	
Protein	33.21
Water	32.54
RMSD	
Bond lengths (Å)	0.003
Bond angles (°)	0.58
Ramachandran plot (%) ^b^	
Most favored regions	98.55%
Allowed regions	1.45%
Disallowed regions	0%

^a^ Values in parentheses are for the highest-resolution shell. ^b^ Ramachandran plot was calculated using the PROCHECK program.

**Table 3 ijms-21-02452-t003:** Thermodynamic parameters for the interaction between various hSNF5^171–253^ and BAF155^SWIRM^ constructs that were obtained by isothermal titration calorimetry at 25 °C.

Description		*K*_D_ (*nM*)	Δ*G* *(kcal/mol)*	Δ*H* *(kcal/mol)*	*−T*Δ*S* *(kcal/mol)*
hSNF5	BAF155^SWIRM^				
171–253 ^a^	Wild-type	110 ± 20	−9.5 ± 0.1	−19.7 ± 0.3	10.2 ± 0.3
174–253 ^a^	Wild-type	110 ± 30	−9.5 ± 0.2	−15.8 ± 0.5	6.3 ± 0.3
179–253 ^a^	Wild-type	300 ± 50	−8.9 ± 0.1	−21.0 ± 0.6	12.1 ± 0.6
181–253 ^a^	Wild-type	300 ± 70	−8.9 ± 0.1	−12.9 ± 0.4	4.0 ± 0.5
183–253 ^a^	Wild-type	290 ± 80	−8.9 ± 0.2	−18.7 ± 0.8	9.8 ± 0.8
186–253 ^a^	Wild-type	760 ± 80	−8.4 ± 0.1	−16.0 ± 0.3	7.6 ± 0.3
171–253/E184A^b^	Wild-type	370 ± 100	−8.8 ± 0.2	−11.2 ± 0.4	2.4 ± 0.4
171–253	N476A/N479A^d^	140 ± 30	−9.3 ± 0.1	−17.7 ± 0.7	8.3 ± 0.3
SFH1/171–253^c^	Wild-type	480 ± 140	−8.6 ± 0.2	−13.4 ± 0.6	4.8 ± 0.6

All the data in this table are derived from ITC experiments. Total free binding energy change ΔG, enthalpy change ΔH, entropy factor change ΔTS and dissociation constant *K*_D_. ^a^
*N*-terminal trucation of hSNF5 mutants. ^b^ E184A mutation of hSNF5^171–253^. ^c^ A chimeric mutant that contains the *N*-terminal tail region from the yeast protein (SFH1). ^d^ N476A and N479A mutations of BAF155^SWIRM^.

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
