# Peer review of "A Coil-to-Helix Transition Serves as a Binding Motif for hSNF5 and BAF155 Interaction"

_ijms, 2020, doi:10.3390/ijms21072452_

Round 1

Reviewer 1 Report

The authors use a combination of X-ray crystallography, NMR spectroscopy and other biophysical method to investigate the molecular association between components of the SWI/SNF complex. Specifically they investigate how the RPT1 domain of SNF5 binds to the SWIRM domain of BAF155. The structure the authors report is distinct form previous structures of this complex because it has an N-terminal disordered region of RPT1 that binds to SWIRM. This importance of this N-terminal helix is confirmed by other experiments. The manuscript is clearly written and present results of general interest. I have a few comments that could improve the quality of the manuscript.

Major:

-The authors state that binding is dramatically reduced (110 nm to 290 nm) upon truncation of the N-terminal helix, but really this is not a huge change given the errors in the measurements. This may suggest that instead of the N-terminal disordered domain ‘locking on’ to the SWIRM domain as observed in the crystal structure, that it instead has a much more transient interaction (which moderately increases the affinity, and helps stabilize the complex) This is supported by the relatively low heteronuclear NOE measurements observed for the sequences that form the α-helix. Consider discussing this.

-The authors state that no structure exists for the human SWI/SNF complex, but do not mention the yeast Cryo-EM structure that was recently published (PMID: 32188938). Although this isn’t a human SWI/SNF complex, it would be interesting to know if the new interaction that the authors outline is conserved in this homologous structure using full-length proteins.

-On lines 107 to 110 the authors state that: “A previous report describing the crystal structure of the hSNF5169-252/BAF155SWIRM complex found that these proteins form a heterotetramer [34], whereas size exclusion chromatography (SEC) and multi-angle light scattering (MALS) data have indicated that hSNF5184-249 is a monomer [35]. To investigate this discrepancy “ To me it is unclear why there is a discrepancy. Is the heterotetramer constructed so that SNF5 would normally be present as a dimer? Some discussion as to why the authors observe a dimer, while previous studies have observed a tetramer could be beneficial. Is the dimerization interface the same between the two structures? Could a crystal contact have been mistaken as a biologically relevant binding interface in the tetramer structure?

Minor:

The authors state on line 190-191 that “We obtained a viable crystal at milder conditions of neutral pH and lower ionic “. , It is unclear what conditions they are comparing their conditions to. Milder than what?

Fig. 5E: A hydrogen bond appears to be annotated between two carbonyl oxygens, this should be corrected.

Author Response

-Major-

Point 1: The authors state that binding is dramatically reduced (110 nm to 290 nm) upon truncation of the N-terminal helix, but really this is not a huge change given the errors in the measurements. This may suggest that instead of the N-terminal disordered domain‘locking on’ to the SWIRM domain as observed in the crystal structure, that it instead has a much more transient interaction (which moderately increases the affinity, and helps stabilize the complex) This is supported by the relatively low heteronuclear NOE measurements observed for the sequences that form the α-helix. Consider discussing this.

Response 1:

We agree with reviewer’s suggestion and included the paragraph of lines 305–310, page 12, as followings:

“When the whole αN helix was removed (hSNF5183–253), the binding affinity reduced by 2.6 fold (KD ~290 ± 80 nM). The moderate decrease in affinity suggests that the αN helix might engage with the BAF155SWIRM via transient interactions. We note that the αN helix exhibited relatively low heteronuclear NOE (XNOE) values even in complex with the BAF155SWIRM domain as shown in Figure 4E. Therefore, the αN helix in the hSNF5171-258/ BAF155SWIRM complex in solution could retain a degree of conformational mobility, providing favorable interactions to stabilize the complex.”

Point 2: The authors state that no structure exists for the human SWI/SNF complex, but do not mention the yeast Cryo-EM structure that was recently published (PMID: 32188938). Although this isn’t a human SWI/SNF complex, it would be interesting to know if the new interaction that the authors outline is conserved in this homologous structure using full-length proteins.

Response 2:

Based on reviewer’s suggestion, we examined the yeast Cryo-EM structure and the structure confirmed that the presence of the αN helix in the BAF complexes. We thus add the following in the Discussion at line 343-346, page 12.

“The yeast Cryo-EM structure showed the presence of αN helix in SWI/SNF complex (ref; Nature https://doi.org/10.1038/s41586-020-2087-1 papers). Further, the hydrogen bond network and hydrophobic interactions between the αN helix and the SWIRM domain in our structure recapitulated, illustrating that the αN helix presents the bona fide interface for the SWIRM domain.”

Point 3: On lines 107 to 110 the authors state that: “A previous report describing the crystal structure of the hSNF5169-252/BAF155 SWIRM complex found that these proteins form a heterotetramer[34], whereas size exclusion chromatography (SEC) and multi-angle light scattering (MALS) data have indicated that hSNF5184-249 is a monomer [35]. To investigate this discrepancy “To me it is unclear why there is a discrepancy. Is the heterotetramer constructed so that SNF5 would normally be present as a dimer? Some discussion as to why the authors observe a dimer, while previous studies have observed a tetramer could be beneficial. Is the dimerization interface the same between the two structures? Could a crystal contact have been mistaken as a biologically relevant binding interface in the tetramer structure?

Response 3:

Yan el al (34) showed that the hSNF5169-252/BAF155SWIRM complex is a heterotetramer based on the analytical gel filtration chromatography and the crystal structure. In our experimental condition, hSNF5 tends to elute earlier than expected from the calculated monomeric molecular weight, so that the gel filtration analysis might mislead the result, showing that the heterotetramer would be formed by dimeric hSNF5. We confirmed it by multi-angle light scattering to produce the absolute molecular weight of hSNF5171-258 (the construct which includes both 169-252 and 184-249 region) pinpointing the monomeric state of hSNF5171-258 in solution. Therefore, we think the crystal structure of the heterotetramer could be attributed by the artefact from crystal contacts.

-Minor-

Point 4: The authors state on line 190-191 that “We obtained a viable crystal at milder conditions of neutral pH and lower ionic “. , It is unclear what conditions they are comparing their conditions to. Milder than what?

Response 4:

Yan et al., made the crystals of hSNF5169-252/BAF155SWIRM complex at buffer containing 0.9 M sodium phosphate monobasic monohydrate and potassium phosphate dibasic, pH 5.4, whereas we obtained the crystals under the buffer containing 15% v/v Tacsimate, pH 7.0, 0.1 M HEPES, pH 7.0, 2% (w/v) polyethylene glycol 3350, which is closer to neutral pH and lower ionic strength. What we meant that we tried to use X-ray crystallization buffer similar to the NMR condition in solution.

Based on reviewer’s comment, we removed “milder” expression to deliver our message clearly (p7, line 191-192)

Point 5: Fig. 5E: A hydrogen bond appears to be annotated between two carbonyl oxygens, this should be corrected.

Response 5:

We appreciate the reviewer’s comment and corrected the hydrogen bonds between two carbonyls as N479 OD1 - E184 N and N479 ND2 - Q182 O in the revised figure 5C.

Reviewer 2 Report

The work describes the SNF5/BAF155 complex part of the chromatin remodeling complex. The structure provides novel and updated information from the N-terminal 171-181 region of SNF5 that transitions from unstructured to helical upon encountering BAF155. The interaction was absent in the previously reported work of Yan L. et.al JMB 2017 (http://dx.doi.org/10.1016/j.jmb.2017.04.008).   While the deletion produces a somewhat modest 3-fold drop in affinity, the melting curves indicate a significant contribution of the N-term helix to the complex stability. In addition, the cancer mutations in the N-term of SNF5 point to a possible biological function that warranted investigation. The manuscript is highly descriptive and comprehensively researched and gives a more complete understanding of the SNF5/BAF155 interaction. Publication is recommended after the points below are addressed.

  • Line 63:  Change  Also, Truncating to Also, truncating.
  • I feel that truncation in the SNF5 N-term should retain Val185. Yes, Glu184 is important for complex formation but removal of Val185 impacts the SNF5 stability by deleting the interaction with Tyr 248 and Met208. More about this below.
  • Figures 4 D the superposition with the 5GJK crystal structure is useful, there is clearly a slight tilt in alpha1 of SNF5, it would be interesting to see if there are difference in relative orientation and sidechain packing between SNF and BAF between this and the previously published crystal structure. I did not see comments about it in the text, maybe I missed it. Panel E how does the TALOS predicted SS from NMR chemical shift match the XNOE and the secondary structure boundaries in the xray structure? Please indicate this on the panel.
  • I find that Figure 5 is hastily designed and conveys almost no direct visual information. This dramatically affects the overall quality of the presentation and the message that is otherwise outstanding by comparison. I strongly advise the authors to remake this figure! Panel C and D are perhaps the most useful, panel A & B are hard to see and of supplementary materials quality, in panel E the centrally located measurement seems to be between to carbonyl oxygens and I am not sure at all what panel F is trying to point to.
  • Fig 6. The 7-fold drop in affinity in the 186-253 SNF5 construct due to E184 removal was already captured in the 5GJ5 structure and does not involve the N-term helix at all. Portraying this fact as the effect of N-term helix binding is misleading. The drop in affinity due to N term is more like 3-fold.
  • Was the effect of residue Pro173 mutant investigated by calorimetry or melting? It seems like an obvious experiment to do. The fact that mutants in the N-term emerge in cancer should be highlighted in the abstract as it may underscore the potential importance of the SNF5 N-term in cancer.  

Author Response

-Major-

Point 1: Line 63:  Change Also, Truncating to Also, truncating

Response 1:

We corrected typo error in Line 63.

Point 2: I feel that truncation in the SNF5 N-term should retain Val185. Yes, Glu184 is important for complex formation but removal of Val185 impacts the SNF5 stability by deleting the interaction with Tyr 248 and Met208. More about this below.

Response 2:

We measured 1H-15N HSQC spectra of hSNF5186-253 to examine its structural stability, but there is no significant chemical shift change except for missing resonances (residue 171-185) and adjacent residues of V185. We believe the decreased affinity is solely due to absence of N-terminal region.

Point 3: Figures 4 D the superposition with the 5GJK crystal structure is useful, there is clearly a slight tilt in alpha1 of SNF5, it would be interesting to see if there are difference in relative orientation and sidechain packing between SNF and BAF between this and the previously published crystal structure. I did not see comments about it in the text, maybe I missed it.

Response 3:

We apologize to have confused the reviewer. Figure 4D shows that superimposed free hSNF5171-258 structure with our complex crystal structure in order to demonstrate structural transition of N-terminal coil to α-helix. We appreciate the construct comment from reviewer and compared our crystal structure with 5GJK. There is no significant difference in relative orientation and side chain packing between two structures.

Point 4: Panel E, how does the TALOS predicted SS from NMR chemical shift match the XNOE and the secondary structure boundaries in the X-ray structure? Please indicate this on the panel.

Response 4:

The annotated secondary structure was obtained from both solution structure of hSNF5171-258 and X-ray structure of hSNF5171-258/BAF155SWIRM, not predicted from NMR chemical shift. We examined the secondary structure boundaries by dihedral angle values between backbone atoms (TALOS, phi and psi).

Point 5: I find that Figure 5 is hastily designed and conveys almost no direct visual information. This dramatically affects the overall quality of the presentation and the message that is otherwise outstanding by comparison. I strongly advise the authors to remake this figure! Panel C and D are perhaps the most useful, panel A & B are hard to see and of supplementary materials quality, in panel E the centrally located measurement seems to be between to carbonyl oxygens and I am not sure at all what panel F is trying to point to.

Response 5:

Thank you for your suggestion. Based on your comments, Figure 5 A and 5B were moved to supplementary figure (Figure S1) and corrected the hydrogen bonds between two carbonyls as N479 OD1 - E184 N and N479 ND2 - Q182 O (Fig. 5C in the revised manuscript). In figure 5F, we observed aliphatic amino acids participating hydrophobic interaction between N-terminal helix of hSNF5171-258 and α2 helix of BAF155SWIRM. However, I agree it would be better to remove it for clear presentation.

Point 6: In Fig 6. The 7-fold drop in affinity in the 186-253 SNF5 construct due to E184 removal was already captured in the 5GJ5 structure and does not involve the N-term helix at all. Portraying this fact as the effect of N-term helix binding is misleading. The drop in affinity due to N term is more like 3-fold.

Response 6:

The affinity with absence of N-terminal helix substantially decreased by 3-fold, however, E184 in the 5GJK could be captured with the presence of N-terminal region that are practically missed in the 5GJK. In our study, we identified the N-terminal helix, and E184 might not easily locate at proper coordination in the absence of N-terminal helix. Therefore, we propose that N-terminal helix not only contributes to binding interfaces between hSNF5 and BAF155 but also plays an important role as an anchor compensating relatively flexible regions.

Point 7: Was the effect of residue Pro173 mutant investigated by calorimetry or melting? It seems like an obvious experiment to do. The fact that mutants in the N-term emerge in cancer should be highlighted in the abstract as it may underscore the potential importance of the SNF5 N-term in cancer.

Response 7:

Thank you for your suggestion. We did not investigate the mutation of Pro173, however, we observed that the deletion mutant hSNF5174–253 did not show any affinity reduction from isothermal calorimetry (ITC) data. As such, we think that Pro173 does not contribute to the binding to the SWIRM domain. Based on your suggestion, we revised the abstract to highlight the potential importance of SNF5 N-terminal in cancer which provides important clue for its function (line 38-39, p1)